# Pulmonary Valve Fibroelastoma, Still a Very Rare Cardiac Tumor: Case Report and Literature Review

**DOI:** 10.3390/diagnostics15030283

**Published:** 2025-01-25

**Authors:** Emanuel-David Anitei, Marius Mihai Harpa, Hussam Al Hussein, Claudiu Ghiragosian, Valentin Ionut Stroe, Paul Calburean, Simona Gurzu, Horatiu Suciu

**Affiliations:** 1Department of Surgery IV, George Emil Palade University of Medicine, Pharmacy, Science and Technology of Targu Mures, 540139 Targu Mures, Romania; anitei_emanuel@yahoo.com (E.-D.A.); alhussein.hussam@yahoo.com (H.A.H.); claudiughiragosian@gmail.com (C.G.); calbureanpaul@gmail.com (P.C.); simonagurzu@yahoo.com (S.G.); horisuciu@gmail.com (H.S.); 2Department of Cardiovascular Surgery, Emergency Institute for Cardiovascular Diseases and Transplantation Targu Mures, 540136 Targu Mures, Romania; valentin.ionut@gmail.com; 3Department of Regenerative Medicine Laboratory, George Emil Palade University of Medicine, Pharmacy, Science and Technology of Targu Mures, 540139 Targu Mures, Romania; 4Research Center of Oncopathology and Translational Medicine (CCOMT), George Emil Palade University of Medicine, Pharmacy, Science and Technology of Targu Mures, 540139 Targu Mures, Romania; 5Department of Pathology, Clinical County Emergency Hospital, 540136 Targu Mures, Romania; 6Department of Medical Research, Romanian Academy of Medical Sciences, 030167 Bucharest, Romania

**Keywords:** cardiac tumor, papillary fibroelastoma, pulmonary valve, soft tissue tumor

## Abstract

**Background and Clinical Significance:** Primary cardiac tumors are among the rarest types of tumor, and until the mid-20th century, they were diagnosed only post-mortem or during other surgical interventions. With the rapid evolution of cardiovascular imaging and the widespread use of echocardiography, the incidence of cardiac fibroelastoma has increased, though it remains one of the rarest primary cardiac tumors. Papillary fibroelastoma is a benign primary cardiac tumor that develops from endocardial tissue, is usually solitary, and can have multiple locations, with the pulmonary valve being one of the rarest sites. The symptoms and complications depend on the tumor’s location, ranging from asymptomatic patients to cerebral ischemic embolism or pulmonary embolism. We analyzed the electronic databases PubMed, Web of Science, and Cochrane and conducted a systematic review of pulmonary valve papillary fibroelastoma (PVPF). Additionally, we included a case from the Adult and Pediatric Cardiovascular Surgery Clinic in Targu Mures. **Case Presentation:** We present the case of a 58-year-old patient who complained of exertional dyspnea. A transthoracic echocardiography (TTE) revealed a tumor mass attached to the pulmonary valve and coronary angiography identified severe coronary lesions. Following discussions within the Heart Team, surgical myocardial revascularization and tumor excision were decided upon due to the thromboembolic risk. Histopathological examination confirmed the diagnosis of papillary fibroelastoma. The postoperative course was uneventful, with an improvement in dyspnea. The mean age of the patients was 60 years, with half being men (*n* = 26, 50%). Regarding symptoms, 34% (*n* = 18) of cases were incidentally identified, while over 30% (*n* = 17) presented with dyspnea. Pulmonary embolism (PE) was reported in only two patients, and the most common associated comorbidities included high blood pressure (HBP) in 33% (*n* = 16) and dyslipidemia in 18%. Tumor size ranged from 0.7 cm to 3 cm with the initial benign cardiac tumor; its occurrence in the pulmonary valve remains exceedingly rare. Due to its frequent overlap with other cardiac pathologies, the clinical presentation is often a nonspecific diagnosis or suspicion of a tumor predominantly established via transthoracic echocardiography in 62% of patients. From a surgical perspective, 63% (*n* = 33) underwent tumor resection with valve sparing, 25% (*n* = 12) required pulmonary valve repair, and three patients necessitated pulmonary valve replacement. **Conclusions:** Although the incidence of papillary fibroelastoma is increasing, making it the most common, there is a need to highlight the indispensable role of echocardiography in diagnosis. Although papillary fibroelastoma is benign, surgical intervention is recommended, particularly in symptomatic patients, or if the tumor exceeds 1 cm in size, exhibits increased mobility, or is present alongside other cardiac surgical procedures.

## 1. Introduction

Primary cardiac tumors are rare, with an incidence of 0.02% in adults, similar to that in the pediatric population [1,2,3]. Advances in the medical field have led to changes in our understanding. Until recently, papillary fibroelastoma was considered the third most common primary cardiac tumor after myxoma and lipoma [4]. However, a comprehensive study in the United States has now identified papillary fibroelastoma as the most frequent benign primary cardiac tumor. Its incidence has increased from 0.019% to 0.089%, with the median age of onset between 60 and 70 years, peaking in the eighth decade. Gender predominance remains difficult to ascertain due to inconsistent findings across studies. It is the most prevalent valvular tumor (88%), with 59% located on the aortic valve, 13% on the mitral valve, 4% on the tricuspid valve, and only 2% on the pulmonary valve [5,6]. The concept that cardiac fibroelastoma is a true neoplasm is not universally accepted, leading to its classification as a non-neoplastic tumor in the 2015 WHO Classification of Tumors of the Heart. In recent years, however, evidence has emerged suggesting that fibroelastoma may indeed be a genuine neoplasm, driven by a KRAS gene mutation, a finding that opens new avenues for research [7,8,9]. We present the case of a 58-year-old male with multiple cardiovascular risk factors and a relevant family history, who presented with exertional dyspnea. Transthoracic echocardiography revealed a tumor measuring approximately 11 mm attached to the pulmonary valve, with an estimated ejection fraction of 55% and no significant wall motion abnormalities. Further investigations, including cardiac MRI, provided precise characterization of the tumor, and coronary angiography identified severe coronary artery disease. The patient subsequently underwent coronary artery bypass grafting and excision of the papillary fibroelastoma, with the histopathological examination confirming the diagnosis. The postoperative course was uneventful.

## 2. Case Report

A 58-year-old male patient with multiple cardiovascular risk factors (high blood pressure, dyslipidemia, peripheral arterial disease, smoker) and a relevant family history (father had a myocardial infarction at age 56) with a history of pituitary adenoma surgery presented with persistent exertional dyspnea. TTE revealed a round, oval tumor attached to the pulmonary valve. The ejection fraction was estimated visually at 55%, with no segmental or global kinetic abnormalities and no signs of pulmonary thromboembolism (PE). The dyspnea persisted; therefore, the patient underwent coronary angiography, which revealed chronic occlusion of the circumflex artery, severe distal stenosis of the right coronary artery, and severe stenosis of the intermediate branch (Figure 1A,B). To accurately assess the tumor formation, the patient underwent cardiac MRI, which described a tumor formation (1.1 × 0.8 cm) as iso-/hypointense on T1-weighted sequences, mildly hyperintense on T2-weighted sequences, with no central enhancement but peripheral enhancement with gadolinium on delayed post-contrast sequences. These tissue characteristics supported the diagnosis of a papillary fibroelastoma (Figure 2A,B). Additionally, no other cardiac tumor structures were identified, and no signs of acute or chronic pulmonary embolism were detected. After obtaining the patient’s written consent and following discussions within the Heart Team, the patient underwent excision of the tumor formation from the pulmonary valve and coronary artery bypass grafting. Intraoperative transesophageal echocardiography (TEE) revealed a round, oval, mobile, well-defined tumor approximately 1.1 × 0.8 cm in diameter, attached to the surface of the right cusp of the pulmonary valve (Figure 1C,D). The tumor did not exert a hemodynamic impact on the valve, was not associated with pulmonary stenosis, and was accompanied by only mild pulmonary regurgitation. No other significant valvular pathologies were observed, and there was no echocardiographic evidence of PE. The surgical intervention was performed via sternotomy, under cardiopulmonary bypass with central cannulation: arterial in the ascending aorta and venous in the right atrium. Cardiac arrest was achieved using Calafiore cardioplegia. Initially, the distal coronary anastomoses were completed, followed by a longitudinal incision of approximately 4 cm in the pulmonary trunk to access the tumor. The tumor was friable, mobile, measuring approximately 1.1 × 1.0 cm, and attached to the pulmonary artery side of the right cusp of the pulmonary valve (Figure 3A). It was completely resected along with a small portion of the right pulmonary cusp that included its implantation base. Since only a small amount of valvular tissue was resected, valve repair was successfully performed using a continuous suture with 5.0 Prolene, Ethicon Inc., Cornelia, GE, USA, without the need for autologous pericardium or other biological materials (Figure 3B–D). Subsequently, the proximal coronary anastomoses were completed. No residual pulmonary regurgitation was detected. No other tumor formations were detected intraoperatively. The macroscopic appearance of the tumor was similar to a sea anemone when placed in serum (Figure 4A). The diagnosis of papillary fibroelastoma was confirmed by histopathological analysis (Figure 4B,C). The postoperative course was favorable and without complications, and the patient was discharged without further complaints of dyspnea.

## 3. Materials and Methods

We conducted a systematic review using the PRISMA 2020 guidelines (Figure 5). To identify and select articles, we utilized the electronic databases PubMed, Web of Science, and Cochrane. We included only human studies, and English-language articles with no publication date restriction. The search terms used were “pulmonary valve fibroelastoma”, “pulmonary papillary fibroelastoma”, and “pulmonic valve fibroelastoma AND cardiac valves fibroelastoma”. We excluded articles not in English, other reviews to avoid duplication of articles, and non-pulmonary valve fibroelastomas. Finally, the selected articles were analyzed and summarized (Table 1).

## 4. Results

We identified *n* = 229 articles in English from the PubMed, Cochrane, and Web of Science databases, with no publication time restrictions. A total of *n* = 57 articles were automatically excluded, and *n* = 4 were manually excluded based on references. Additionally, *n* = 7 articles were excluded based on study design and settings, and one article (*n* = 1) was excluded after careful analysis because the tumor was not located on the pulmonary valve but adjacent to it. In the end, *n* = 51 articles were included and thoroughly analyzed, summarized in the table above, to which the case from our clinic was also added. All 51 articles are case reports, with two of them including two cases each. In total, *n* = 54 cases of pulmonary valve fibroelastoma were summarized.

The mean age of the patients was 60 years, with a range between 30 and 85 years. Half were men, *n* = 26 (50%), while women, *n* = 23, represented 45%, and in three cases, *n* = 3, the sex was unknown. Regarding symptoms, 34% (*n* = 18) were incidentally identified, with 11% of them having other cardiac pathologies for which they underwent concomitant surgery. In over 30% of cases (*n* = 17), patients presented with dyspnea, while atypical chest pain was present in 17% of the patients; syncope and palpitations were also present in 7% and 5% of cases, respectively. The most common associated pathologies were HBP, present in 33% (*n* = 16) of patients, followed by dyslipidemia (18%), diabetes (12%), PAD (8%), and PE (4%). Tumor size ranged from 0.7 cm to 3 cm. The initial diagnosis or suspicion of a tumor formation was made through transthoracic echocardiography in the majority of patients (62%). In 20% of patients, the initial diagnosis was made through CT, noting that most of these diagnoses were made in symptomatic patients. The fibroelastoma was initially detected through MRI in a single case. The diagnosis of papillary fibroelastoma was made through histopathological examination. No cases of pulmonary valve stenosis were recorded; however, one case presented with severe pulmonary insufficiency due to the tumor involving all three cusps, which led to pulmonary valve replacement. Three cases of moderate insufficiency were also recorded. As for the outcome, 98% of patients had an uneventful postoperative course. Two patients had pulmonary thromboembolism, and *n* = 6 patients were treated with anticoagulants. In *n* = 25 patients (56%), the surgical indication was tumor mobility, while in 27% of patients, the indication was concomitant surgery. Two other cases had surgery at the patient’s request, and two cases did not have a surgical indication due to the high surgical risk, so they received anticoagulant treatment. Regarding the surgical strategy, 63% (*n* = 33) of patients underwent tumor resection with valve sparing, 25% (*n* = 12) required pulmonary valve repair, and three patients needed pulmonary valve replacement. One patient underwent aspiration of the tumor formation using the AngioVac system, AngioDynamics, New York, USA. Two patients underwent minimally invasive surgery through left anterolateral thoracotomy for the removal of the fibroelastoma, with valve preservation in both cases. One patient had the tumor removed on a beating heart, with the valve also spared in this case. In some cases requiring repair, bovine pericardium was used, while in others, autologous pericardium was utilized.

In 31% of cases, the location of the fibroelastoma is either unknown or not clearly identified. The most affected cusp is the right cusp, present in 26% of cases (Figure 6). Additionally, the most common location of the fibroelastoma is on the ventricular surface, found in *n* = 29 (54%) of cases, as you can see in Figure 7.

## 5. Discussion

### From Historical to New Perspectives

Papillary fibroelastoma, a type of benign cardiac tumor, has been extensively studied over the decades, with several significant medical discoveries. In 1931, Yaters described valvular tumors for the first time, and in 1934, Campbell and Carling associated sudden death with a valvular tumor. The term “papillary fibroelastoma” was first used by Cheitlin and colleagues in 1975, and in the same year, Fishbein and colleagues analyzed and described fibroelastoma using electron microscopy. In 1977, Hugh A. McAllister and colleagues identified papillary fibroelastoma as the third most common benign primary cardiac tumor, and Anderson and colleagues described the first case of congenital papillary fibroelastoma [6]. In 1979, Lichtenstein and colleagues incidentally discovered fibroelastoma during surgery for the closure of a ventricular septal defect. Flotte and colleagues echocardiographically described fibroelastoma for the first time in 1980 [61]. In 1998, Speights and colleagues published the first cytogenetic studies of papillary fibroelastoma [62]. In 2003, Ramesh M. Gowda and colleagues recognized papillary fibroelastoma as the second most common benign primary cardiac tumor [6]. In 2015, fibroelastoma was classified as a non-neoplastic tumor by the WHO Classification of Tumors of the Heart, but in the same year, according to Syahidah S. Tamin, it became the most common benign primary cardiac tumor [5,9]. Between 2017 and 2020, the first evidence of the oncogenic etiology (mutations in the KRAS gene) of papillary fibroelastoma was presented by Maike Wittersheime and Melanie C. Bois and their colleagues [7,8]. In 2021, papillary fibroelastoma was officially designated as a neoplastic tumor (ICD-O 8898/0) in the WHO Classification of Tumors of the Heart (2021) [63].

Primary cardiac tumors are rare, with a meta-analysis of 731,309 autopsies revealing an incidence of 0.02%, a rate similar to that observed in the pediatric population [1,2,3]. In the past, papillary fibroelastoma was discovered incidentally during autopsy or intraoperatively during other cardiac surgeries. Its incidence has been increasing, from 0.019% to 0.089%, primarily due to the widespread use of imaging modalities such as echocardiography, CT, and MRI. Papillary fibroelastoma is the most common primary cardiac tumor, surpassing myxoma and lipoma. It is the most frequent valvular tumor, with 88% of cases located on valves, and the majority (59%) found on the aortic valve. The pulmonary valve remains the least affected, with only 2% of cases [5]. The average age of onset is between 60 and 70 years, with the highest risk at 80 years of age, predominantly affecting males, although some studies show a female predominance. In our study, the distribution was approximately equal [5,6,64]. Fibroelastomas can occur anywhere in the heart, with symptoms and complications closely related to their location. Those on the left side of the heart can lead to devastating complications such as ischemic stroke, angina, myocardial infarction, peripheral infarcts and emboli, and even sudden death. Those on the right side of the heart are often asymptomatic but can cause pulmonary thromboembolism, which may lead to pulmonary hypertension or paradoxical emboli in the case of intracardiac communications. In our study, 34% of patients with pulmonary valve fibroelastomas were asymptomatic. The majority of patients (62%) were diagnosed through transthoracic echocardiography, which is an indispensable and widely used investigation [65]. Dyspnea is a common symptom in cardiac pathology, appearing in 32% of patients in our study. In the context of cardiac tumors such as papillary fibroelastoma, it is associated with the size of the tumor and its impact on valvular function or hemodynamics. Considering that the tumor does not interfere with valvular function and does not cause hemodynamic alterations in the right ventricular outflow tract, it is less likely that the dyspnea was caused by the tumor. However, due to its overlap with coronary artery disease, it is difficult to draw a definitive conclusion, as dyspnea could potentially result from both pathologies. Notably, following tumor excision and surgical revascularization, the dyspnea resolved [60]. Other atypical symptoms may include fever and thrombocytopenia [2]. While the infectious risk associated with cardiac myxoma is well-documented, there is limited information regarding the infectious risk of papillary fibroelastoma. However, a recent case report describes an infected papillary fibroelastoma that resulted in pulmonary embolization [66,67]. In the effort to determine the etiology of papillary fibroelastoma, its potential association with a true syndrome has been considered. In our patient’s case, this possibility is plausible given the history of surgical excision of a pituitary adenoma, though more extensive studies are required to validate this theory [68]. Previous studies have suggested that papillary fibroelastoma may be a stress-induced tumor, as it most commonly occurs in areas of increased hemodynamic stress or following endocardial injury from cardiac interventions or radiation. A viral etiology, particularly Cytomegalovirus, has also been proposed, along with associations with rheumatic disease and the aging process, as the highest incidence occurs around 80 years of age [69]. Despite the 2015 WHO Classification of Tumors of the Heart categorizing papillary fibroelastoma as a non-neoplastic entity, a study by Mike W. et al. demonstrated that 79% of the analyzed specimens exhibited a KRAS gene mutation [7,9]. Similarly, another study involving 50 cases of papillary fibroelastoma confirmed this finding, with a KRAS mutation present in over 30% of cases [8]. These studies provide substantial evidence supporting the oncogenic origin of papillary fibroelastoma, at least in certain subtypes. As a result, the 2021 WHO Classification of Tumors of the Heart reclassified it as a genuine neoplastic tumor driven by an oncogenic driver mutation [63]. The thromboembolic potential of papillary fibroelastoma is attributed to its histological architecture, where fibrillar extensions facilitate thrombus formation, and the inherent fragility of these structures predisposes them to fragmentation and subsequent embolization [69]. The mobility and embolic potential of papillary fibroelastoma are significant risk factors, with the incidence of embolic events being 53% higher compared to atrial myxomas [70]. Tumor mobility was the only independent factor significantly associated with papillary fibroelastoma-related death or nonfatal embolization (*p* = 0.001) [5,17,37]. Given that papillary fibroelastoma does not infiltrate or destroy valvular tissue, surgical treatment should be performed in centers experienced in valve repair. Indications for surgical intervention include symptomatic patients, those requiring concomitant cardiac procedures, asymptomatic individuals with tumors larger than 1 cm or exhibiting significant mobility, and tumors located on the left side of the heart or the right side with associated congenital heart defects (e.g., atrial septal defect, ventricular septal defect) [5,6,32,71,72]. In the present study, pulmonary valve sparing, either through shave resection or repair, represents the primary surgical strategy, although cases are described where the valve is so severely affected that sparing is not feasible. Given that the tumor was greater than 1 cm, exhibited mobility with an elevated risk of embolization, and the patient was scheduled for coronary artery bypass grafting, surgical excision was deemed necessary. For patients who are not suitable candidates for surgical removal, antiplatelet or anticoagulant therapy is recommended. A tumor aspiration procedure using a percutaneous system is described, which was used in some patients. However, larger studies are needed to ensure the safety and effectiveness of this procedure [5,59,66,70]. While previous studies have not documented recurrence of papillary fibroelastoma, research by Syahidah S. Tamin et al. from the Mayo Clinic (Rochester, Minnesota) reports a postoperative recurrence rate of 1.6%. Most tumors were initially diagnosed via echocardiography, which has limitations in detecting small-sized formations and depends on the examiner’s experience and skill. A possible explanation for recurrence could be the initial failure to identify small, concomitant tumors that have since grown and become visible over time [5,18,73]. The differential diagnosis includes both benign tumors, such as atrial myxoma, Lambl’s excrescences, infectious endocarditis vegetations, lipoma, rhabdomyoma, leiomyoma, and lymphangioma, as well as malignant tumors, including angiosarcoma, mesenchymoma, leiomyosarcoma, fibromyxosarcoma, and metastatic lesions. All these entities were excluded through imaging studies and histopathological examination [18,60,74,75].

## 6. Conclusions

Papillary fibroelastoma has surpassed both myxomas and lipomas, making it the most common primary cardiac tumor. Due to advances in genetic research, it is now classified as a true tumor, associated with mutations in the KRAS gene. Patients with pulmonary valve papillary fibroelastoma are typically asymptomatic, with the tumor most often discovered incidentally. Echocardiography should be considered an essential diagnostic tool for its detection. Surgical intervention is considered the treatment of choice for symptomatic patients, for tumors larger than 1 cm with high mobility and an increased risk of thromboembolism, and for asymptomatic patients undergoing concomitant surgical procedures. As an alternative, anticoagulant therapy is recommended for isolated cases not amenable to surgical treatment. Additionally, certain percutaneous aspiration systems have shown promise in selected cases.

## Figures and Tables

**Figure 1 diagnostics-15-00283-f001:**
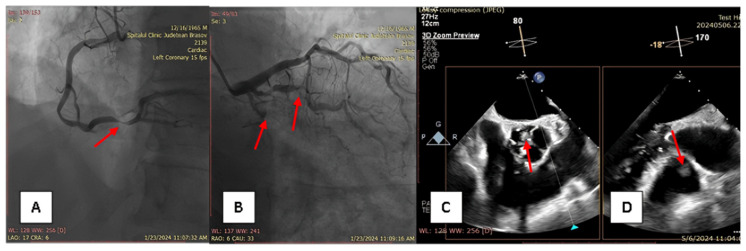
(**A**)—severe stenosis of right coronary artery; (**B**)—chronic occlusion of circumflex artery and severe stenosis of intermedius branch; (**C**,**D**)—transesophageal echocardiography, showing papillary fibroelastoma attached to the right cusp of the pulmonary valve.

**Figure 2 diagnostics-15-00283-f002:**
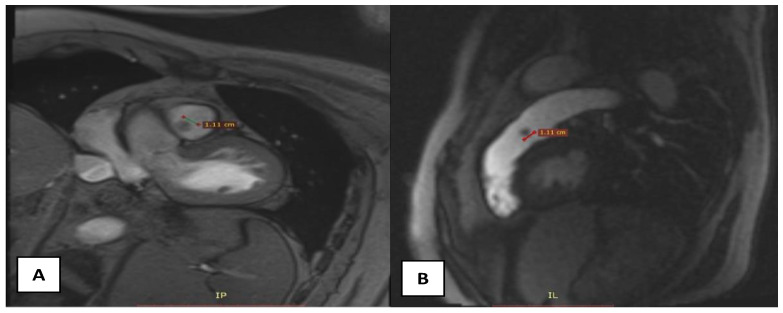
(**A**)—transversal section on IRM demonstrating a 1.1 × 0.8 cm spherical structure attached to the pulmonary valve; (**B**)—sagittal section on IRM.

**Figure 3 diagnostics-15-00283-f003:**
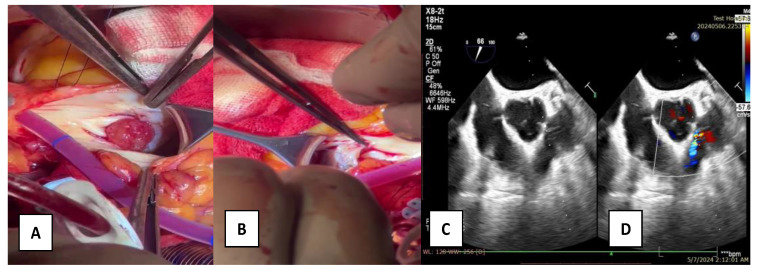
(**A**)—intraoperative aspect of the papillary fibroelastoma; (**B**)—repair of the right cusp of the pulmonary valve; (**C**,**D**)—transesophageal echocardiography, postoperative aspect with no residual regurgitation.

**Figure 4 diagnostics-15-00283-f004:**
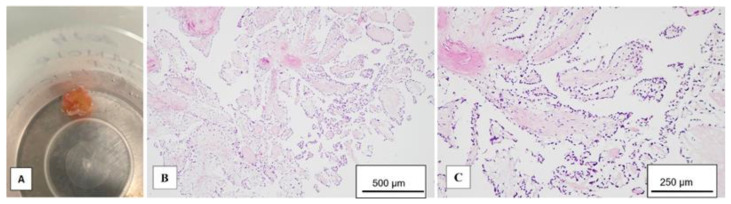
The postoperative characteristics of the papillary fibroelastoma. (**A**)—the “sea anemone” macroscopic aspect; (**B**,**C**)—microscopic examination reveals multiple avascular villous projections (**B**) covered by endothelial cells (**C**).

**Figure 5 diagnostics-15-00283-f005:**
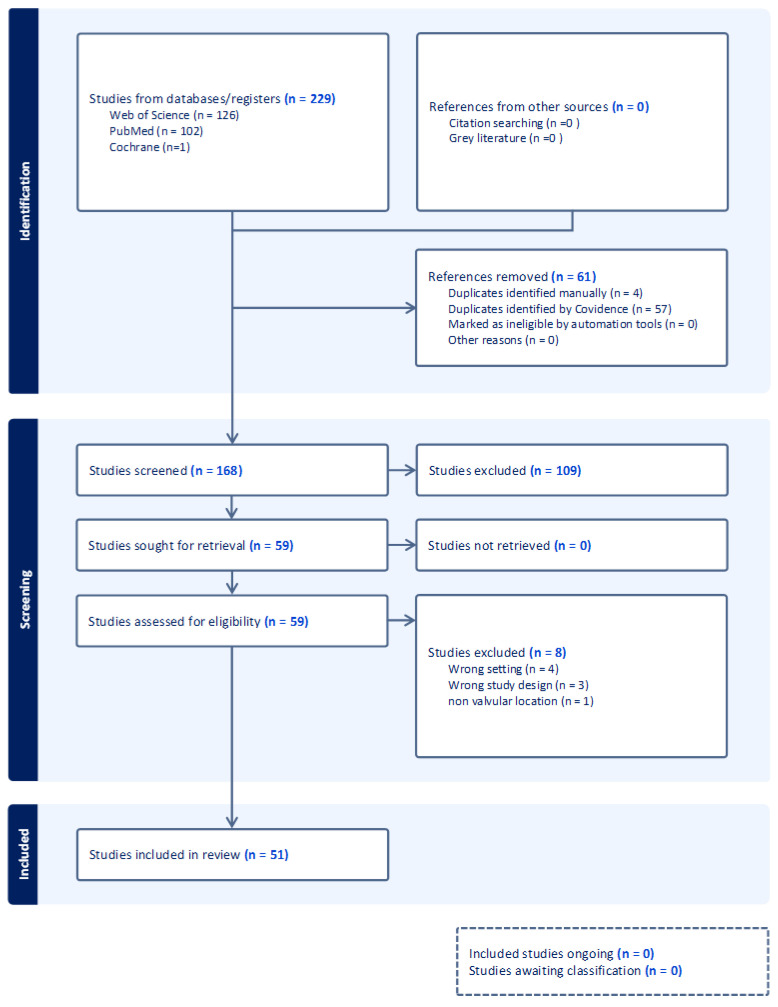
PRISMA flow diagram.

**Figure 6 diagnostics-15-00283-f006:**
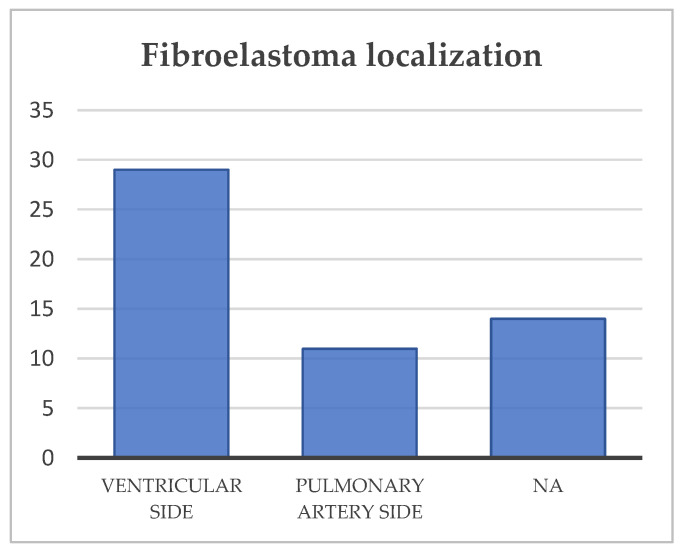
Tumor localization. NA—not available.

**Figure 7 diagnostics-15-00283-f007:**
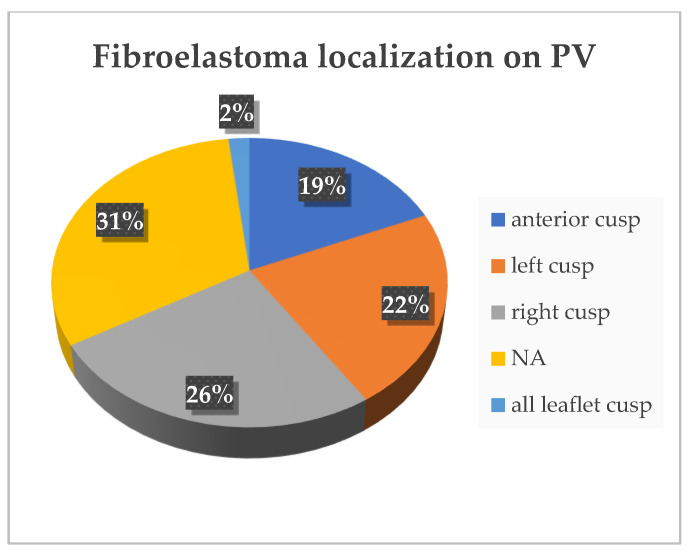
Tumor localization on PV. NA—not available.

**Table 1 diagnostics-15-00283-t001:** Summary of results.

Author	Age	Sex	Symptoms	Other Cardiac Diseases	Associated Conditions	TTE (Mass cm)	TEE (Mass cm)	CT Scan (Mass cm)	MRI (Mass cm)	Surgery Indication	PV Repair/Replacement
A. Singh et al. [10]	40	M	ACP	no	no	* 1.8 × 1.4	no	no	no	size	TU resection
T. Grus et al. [11]	54	M	DOE	no	Hodgkin lymphoma	yes	yes	* 16 mm	no	size/mobility	repair
J.C. George et al. [12]	76	M	dyspnea, fatigue	MR, iCAD	no	no	1.2 × 1.1	no	yes *	concomitant surgery	repair
A.Yamashita et al. [13]	45	M	palpitations	no	no	yes *	no	no	no	NA	repair
J.L. Lopes et al. [14]	NA	NA	IF	no	no	yes *	yes	no	no	NA	repair
J.M. Tomasko et al. [15]	49	F	IF	aortic valve PF	no	yes *	yes	yes	yes	concomitant surgery	repair
S. Singireddy et al. [16]	49	F	ACP, dyspnea, postural hypotension	no	HBP	yes	no	yes *	0.9 × 1.5	mobility/symptomatic	TU resection
H. Uehara et al. [17]	70	F	IF	AR	HBP, Dyslipidemia	* 1.9 × 1.5	yes 1.6	1.7 × 1.4 × 0.8	yes	concomitant surgery	TU resection
A. Molnar et al. [18]	55	F	DOE	no	HBP, Dyslipidemia	yes	* 1.0 × 1.0	yes	0.7	mobility/risk of embolism	TU resection
M.J. Costa et al. [19]	75	M	dyspnea	iCAD, PCI	HBP, Dyslipidemia, PAD, hypothyroidism	yes *	1.3 × 1.9	no	no	size	TU resection
D. Guo et al. [20]	51	M	dyspnea, syncope, cough, sputum	no	asthma, respiratory tract infection	* 1.6 × 1.0	yes	yes	1.3 × 1.0 × 0.9	mobility/risk of embolism	TU resection
J.F Val-Bernal et al. [21]	60	M	IF	aortic valve PF	no	* 2.5 × 1.5	no	no	no	size/mobility	TU resection
S. Lee et al. [22]	43	M	ACP	no	no	* 1.4 × 1.3	yes	yes	no	NA	TU resection
S. Hajouli et al. [23]	65	M	IF	iCAD, PCI, SVT	HBP, Dyslipidemia, PAD, COPD	yes	yes *	no	no	concomitant surgery	TU resection
S. Biočić et al. [24]	32	F	palpitations	no	no	yes *	1.1 × 1.0	no	no	mobility	repair
N.S Bhagwandien et al. [25]	42	F	ACP	NA	NA	* 1.0 × 0.5	yes	no	no	mobilty	TU resection
M. Kovacevic et al. [26]	64	F	IF	iCAD	no	* 1.7 × 1.3	yes	no	no	concomitant surgery	TU resection
A.C. Yopp et al. [27]	68	F	dyspnea	no	no	yes 2.8	no	* 3.0	no	size/mobility	TU resection
G.H. Yao et al. [28]	49	M	atypical symptoms	no	no	yes *	2.5 × 2.0	no	no	size/mobility	TU resection
K. Okada et al. [29]	71	F	DOE, fever	no	no	no	yes	* 2.0	no	size/mobility	TU resection
M.H. van Werkum et al. [30]	64	M	fatigue, bradicardia	no	HBP, Ménière’s disease	yes *	1		1.0 × 0.7	mobility	TU resection
D.K. Hosseini [31]	53	F	dyspnea, ACP	NA	mediastinal tumor	yes	yes *	yes	yes	NA	NA
M.Y. Park et al. [32]	43	M	ACP	no	HBP, Dyslipidemia, diabetes	* 1.3 × 0.7	yes	yes	no	mobility/risk of embolism	TU resection
D.L. Ngaage et al. [33]	NA	NA	NA	NA	NA	yes *	NA	NA	NA	NA	valve replacement
D.L. Ngaage et al. [33]	NA	NA	NA	NA	NA	yes *	NA	NA	NA	NA	NA
D. Jilani et al. [34]	83	F	DOE	iCAD, AFIB, permanent pacemaker	HBP, PHT, PE, sleep apnea	* 1.5 × 1.4	no	yes	no	anticoagulation	no
O. Siddiqui et al. [35]	64	F	hemoptysis, acute respiratory failure	no	nephrolithiasis	no	1.21 × 1.07	yes *	yes	mobility/risk of embolism	TU resection
M. Uchino et al. [36]	66	F	IF	ascending aorta and aortic valve disease	NA	* 1.3	NA	NA	NA	concomitant surgery	valve replacement
T. Generali et al. [37]	56	M	IF	no	no	* 1.3 × 0.9	yes	no	yes	mobility/risk of embolism	TU resection
S. Tobe et al. [38]	73	M	IF	no	hepatocellular carcinoma	* 2.6 × 2.1	yes	no	no	mobility/risk of embolism	repair
M. Cecconi et al. [39]	75	F	effort angina	iCAD	diabetes	* 1.5	yes	no	no	concomitant surgery	TU resection
S.S. Vittala et al. [40]	53	F	IF	AR, tricuspid and mitral PF	no	yes *	yes	no	no	concomitant surgery	TU resection
D.F. Sanfeliu et al. [41]	30	F	IF	no	no	yes 1.5	yes	no	yes	mobilty/risk of embolism	repair
J.R. Nellis et al. [42]	53	F	palpitations, angina, tahicardia, syncope	no	no	yes *	yes	NA	1	NA	repair
A.A. Rahsepar et al. [43]	48	F	pre-syncope	no	no	* 0.8 × 0.8	yes	yes	1.6 × 1.0 × 0.8	mobility/risk of embolism	repair
L. Banuls et al. [44]	74	M	syncope	iCAD	stroke, lung nodule, HBP, dyslipidemia, diabetes	* 1.3 × 1.1	1.2 × 1.1	no	no	concomitant surgery	TU resection
A. G. Iosifescu et al. [45]	62	F	no	mitral, tricuspid PF	stroke	yes * 0.7	yes	no	no	mobility/risk of embolism	TU resection
S. Ahern et al. [46]	67	J	ACP	no	no	yes	* 1.82 × 1.35	no	yes	size/mobility/risk of embolism	TU resection
M. Daccarett et al. [47]	52	M	IF	no	HBP, Dyslipidmia	yes *	1.5 × 1.4	no	no	size/mobility/risk of embolism	TU resection
C. Gustafson et al. [48]	81	M	IF	iCAD	PAD, chronic emphysema	no	* 0.4	no	no	anticoagulation	no
C. Gustafson et al. [48]	81	F	dyspnea	AFIB	PE	no	* 1.0	no	no	mobilty/risk of embolism	TU resection
P. Fonseca et al. [49]	42	M	IF	no	HBP, diabetes	yes *	0.8 × 0.7	yes	no	mobility/risk of embolism	TU resection
A. Mete et al. [50]	72	M	DOE	no	HBP, Dyslipidemia	yes *	NA	NA	yes	mobility/risk of embolism	repair
F. Kirk et al. [51]	52	F	ACP, dyspnea	no	alcoholic pancreatitis, diabetes	yes	1.3 × 1.0	yes *	no	mobility/risk of embolism	TU resection
F. Annie et al. [52]	70	M	IF	iCAD	no	no	1.4 × 0.9	yes *	yes	concomitant surgery	TU resection
C. Jellis et al. [53]	67	F	DOE	iCAD	HBP, diabetes, nephropathy, retinopathy	yes	no	* 1.0	no	NA	repair
DiLorenzo WR et al. [54]	85	M	syncope, fatigue	NA	NA	yes *	0.8 × 0.8	no	no	patient request	TU resection
M. Ibrahim et al. [55]	60	F	IF	no	HBP, leg melanoma	1.4 × 1.0	yes	yes	no	patient request	TU resection
A. Teis et al. [56]	45	M	IF	no	Crohn’s disease	yes	yes	yes *	1.2	NA	TU resection
D. Papasaikas et al. [57]	70	M	DOE	no	no	yes *	2.2 × 1.6	no	no	size/mobility/risk of embolism	TU resection
H. Yagoub et al. [58]	74	M	IF	iCAD	NA	* 1 × 1	no	no	no	concomitant surgery	TU resection
J.J. Alexis et al. [59]	64	M	IF	permanent pacemaker	HBP, diabetes, diabetes, sacral ulcer, sepsis	yes	0.8 × 0.7	yes *	no	angiovac	TU aspiration
J. Taylor et al. [60]	67	F	DOE, ACP, bilateral lower extremity edema	no	Dyslipidemia	yes	1.3 × 1.3	no	1.1 × 1.0	NA	valve replacement
Presented Case	58	M	dyspnea	iCAD	HBP, Dyslipidemia, PAD, pituitary adenoma	yes *	1.1 × 0.8	no	1.1 × 0.8	concomitant surgery	repair

Legend: TTE—transthoracic echocardiography; TEE—transesophageal echocardiography; CT—computed tomography; MRI—magnetic resonance imaging; F—female; M—male; IF—incidental finding; DOE—dyspnea on exertion; ACP—atypical chest pain; AR—aortic regurgitation; MR—mitral regurgitation; iCAD—ischemic coronary artery disease; PAD—peripheral artery disease; HPB—high blood pressure; NA—not available; TU—tumor; AFIB—atrial fibrillation; PF—fibroelastoma; PV—pulmonary valve; SVT—supraventricular tachycardia; PCI—percutaneous coronary intervention; PE—pulmonary embolism; PHT—pulmonary hypertension; COPD—chronic obstructive pulmonary disease. * Indicates the investigation that initially detected the fibroelastoma.

## Data Availability

Data Availability Statements are available in the Emergency Institute for Cardiovascular Diseases and Transplantation Targu Mures database and can be requested from the corresponding author.

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
