# Peer review of "Pulmonary Valve Fibroelastoma, Still a Very Rare Cardiac Tumor: Case Report and Literature Review"

_diagnostics, 2025, doi:10.3390/diagnostics15030283_

Round 1

Reviewer 1 Report

Comments and Suggestions for Authors

The author presents a case report and a literature review on fibroelastoma. The paper is well written and follows a logical thread.

The first part of the manuscript describes a patient with coronary heart disease who required surgery. Incidentally a tumor adhering to the pulmonary valve was discovered. The patient underwent coronary artery bypass surgery. Although there was no impairment of pulmonary valve function or hemodynamics, the fibroelastoma was removed at the same time due to the high risk of thromboembolism. The surgical outcome was excellent.

The second part of the manuscript is a systematic review. The author reviews 51 articles on pulmonary valve fibroelastoma, all of them being case reports. Although fibroelastoma is a common cardiac tumor the incidence in the general population is very low (0,02%). Furthermore the pulmonary valve is rarely affected. I personally only found fibroelastoma on the aortic valve. Treatment consisted of either removing the tumor and then shaving the affected area or resection and repair with CardioCel. This part of the manuscript and the following discussion are appropriate and express ideas clearly and well argued.

I have a few remarks which have to be addressed before publication:

1.       Line 74-78: Transthoracic echocardiography is usually abbreviated– TTE and transesophageal echocardiography is usually abbreviated – TEE. Neither the TTE nor the TEE describe the function of all valves and especially the pulmonary valve. Although, echocardiography plays a key role in diagnosis of pulmonary embolism(PE) either CT (the gold standard) or MRI can diagnose or exclude acute or chronic PE. Which is the case here?

2.       Line 91 - 95: This being a case report it should describe in detail the surgical management. For certain median sternotomy, cardiopulmonary bypass with bicaval cannulation?, cardioplegia?, Incision on the pulmonary artery?, and most importantly how was  the tumor surgically approached. The author describes that a segment of the cusp of the pulmonary valve was excised but how did the surgeon repair it afterwards? Did the CABG follow the treatment of the fibroelastoma or the other way around?

3.       Line 93: “superior surface of the right cusp” – either ventricular side or pulmonary artery side. It may confuse the reader otherwise. The author uses it correctly in Figure 6.

4.       Line 259 – No changes have to be made. I strongly agree with the authors argument.

Reviewer 2 Report

Comments and Suggestions for Authors

This case report showed the case of a 58-year-old male with multiple cardiovascular risk factors and a relevant family history, who presented with exertional dyspnea. After the diagnosis by transthoracic echocardiography, which revealed a tumor measuring approximately 11 mm attached to the pulmonary valve, with an estimated ejection fraction of 55% and no significant wall motion abnormalities, cardiac MRI, and coronary angiography, the patient subsequently underwent coronary artery bypass grafting and excision of the papillary fibroelastoma. The histopathological examination confirmed the diagnosis of Pulmonary valve fibroelastoma, and the postoperative course was uneventful.  

The reviewer acknowledges the clinical significance of the present case report and has provided the following comments: 

Major comments: 

1.       In Figure 1, panels C and D are referenced before panels A and B in the text. It would be preferable to rearrange the figure panels (A–D) to follow the order in which they are discussed in the text. 

2.       The patient reported dyspnea, which improved after surgical treatment. Could the authors provide information on the trends of BNP (or NT-proBNP) levels and exercise tolerance (e.g., results of a 6-minute walk test) before and after the surgical intervention? 

Minor comment: 

3.       Line 226: There is a typographical error with a double period ("..") following the word "hemodynamics." This should be corrected. 

Round 2

Reviewer 1 Report

Comments and Suggestions for Authors

The author presents a case report and a literature review on fibroelastoma. The paper is well written and follows a logical thread.

The first part of the manuscript describes a patient with coronary heart disease who required surgery. Incidentally a tumor adhering to the pulmonary valve was discovered. The patient underwent coronary artery bypass surgery. Although there was no impairment of pulmonary valve function or hemodynamics, the fibroelastoma was removed at the same time due to the high risk of thromboembolism. The surgical outcome was excellent.

The second part of the manuscript is a systematic review. The author reviews 51 articles on pulmonary valve fibroelastoma, all of them being case reports. Although fibroelastoma is a common cardiac tumor the incidence in the general population is very low (0,02%). Furthermore the pulmonary valve is rarely affected. I personally only found fibroelastoma on the aortic valve. Treatment consisted of either removing the tumor and then shaving the affected area or resection and repair with CardioCel. This part of the manuscript and the following discussion are appropriate and express ideas clearly and well argued.

I congratulate the authors for a well written and interesting paper.